# IDEAL: Query-Efficient Data-Free Learning from Black-box Models

**Jie Zhang** [*†]
Zhejiang University
{zj_zhangjie}@zju.edu.cn

**Chen Chen**[*] **& Lingjuan Lyu**[‡]
Sony AI
{ChenA.Chen,Lingjuan.Lv}@sony.com

## Abstract

Knowledge Distillation (KD) is a typical method for training a lightweight student model with the help of a well-trained teacher model. However, most KD methods require access to either the teacher's training data or model parameter, which is unrealistic. To tackle this problem, recent works study KD under data-free and black-box settings. Nevertheless, these works require a large number of queries to the teacher model, which incurs significant monetary and computational costs. To address these problems, we propose a novel method called *query-effIcient Data-free lEarning from blAck-box modeLs* (IDEAL), which aims to query-efficiently learn from black-box model APIs to train a good student without any real data. In detail, IDEAL trains the student model in two stages: data generation and model distillation. Note that IDEAL does not require any query in the data generation stage and queries the teacher only once for each sample in the distillation stage. Extensive experiments on various real-world datasets show the effectiveness of the proposed IDEAL. For instance, IDEAL can improve the performance of the best baseline method DFME by 5.83% on CIFAR10 dataset with only $0.02\times$ the query budget of DFME.

## 1 Introduction

Knowledge Distillation (KD) has emerged as a popular paradigm for model compression and knowledge transfer Gou et al. (2021). The goal of KD is to train a lightweight student model with the help of a well-trained teacher model. Then, the lightweight student model can be easily deployed to resource-limited edge devices such as mobile phones. In recent years, KD has attracted significant attention from various research communities, e.g., computer vision Wang (2021); Passalis et al. (2020); Hou et al. (2020); Li et al. (2020), natural language processing Hinton et al. (2015); Mun et al. (2018); Nakashole & Flauger (2017); Zhou et al. (2020b), and recommendation systems Kang et al. (2020); Wang et al. (2021a); Kweon et al. (2021); Shen et al. (2021).

However, most KD methods are based on several unrealistic assumptions: (1) users can directly access teacher's training data; (2) the teacher model is considered as a white-box model, *i.e.*, model parameters and structure information can be fully utilized. For example, to facilitate the training process, FitNets Romero et al. (2015) uses not only the original training data, but also the output information from the teacher's intermediate layers. However, in real-world applications, the teacher model is usually provided by a third party. Thus the teacher's training data is usually not public and unable to access. In fact, the teacher model is mostly trained by big companies with extensive amounts of data and plenty of computation resources, which is the core competitiveness of companies. As a result, the specific parameters and structural information of the teacher model are never exposed in the real world. Consequently, accessing the teacher model or teacher's training data render these KD methods impractical in reality.

To solve these problems, some recent studies Truong et al. (2021b); Fang et al. (2021a) attempt to learn from a black-box teacher model without any real data, i.e., data-free black-box KD. These

---

[*]Equal contribution.
[†]Work done during internship at Sony AI.
[‡]Corresponding author.

Table 1: An empirical study of previous methods with a limited number of queries (we set the query budget $Q = 25K$ for MNIST, $Q = 250K$ for CIFAR10, and $Q = 2M$ for CIFAR100.) in various scenarios. We also adopt CMI Fang et al. (2021b) for hard-label scenarios and name it "CMI*".

| Method | access to training data | white-box / black-box | logits / hard-label | MNIST | CIFAR10 | CIFAR100 |
|---|---|---|---|---|---|---|
| Normal KD Hinton et al. (2015) | ✓ | white-box | logits | 98.91% | 94.34% | 76.87% |
| CMI Fang et al. (2021b) | × | white-box | logits | 98.20% | 92.22% | 74.47% |
| CMI* | × | white-box | hard-label | 86.53% | 76.17% | 63.45% |
| DFME Truong et al. (2021a) | × | black-box | logits | 68.26% | 51.28% | 39.12% |
| ZSDB3 Wang (2021) | × | black-box | hard-label | 37.33% | 32.18% | 14.28% |

methods do not need to access the private data and can train the student model with the **class probabilities** returned by the teacher model. However, in real-world scenarios, the pre-trained model on the remote server may only provide APIs for inference purpose (*e.g.*, commercial cloud services), these APIs usually return the top-1 class (*i.e.*, **hard label**) of the given queries. For example, Google BigQuery[1] provides APIs for several applications. Such APIs only return a category index for each sample instead of the class probabilities. Moreover, these APIs usually charge for each query to the teacher model, and thus budget should be considered in the process of query. Nevertheless, previous methods Truong et al. (2021a); Wang (2021); Zhou et al. (2020a) require **a large number of queries** to the teacher model, which is costly and impractical. Hence, training a high-performance student model with a small number of queries is still an unsolved problem.

In this paper, we consider a more practical and challenging setting: (1) the teacher's training data is not accessible, *i.e.*, *data-free*; (2) the parameter of the teacher model is not accessible, *i.e.*, *black-box*; (3) the teacher model only returns a category index for each sample, *i.e.*, *hard-label*; and (4) the number of queries is limited, *i.e.*, *query-efficient*. To better understand the difficulty of this setting, we report the top-1 test accuracy of student models under different scenarios with a limited query budget[2] in Table 1.

As shown in Table 1, we have some valuable observations: (1) In white-box scenarios, data-free KD can achieve satisfied performance, but when the model API is restricted to only hard labels, CMI Fang et al. (2021b) suffers from serious performance degradation. It indicates that logits can provide more information for training, while hard labels are more difficult; (2) With the same number of queries, the performance of these methods dramatically decrease under the black-box scenarios. Furthermore, the performance of data-free black-box KD with hard labels is only 14.28% on CIFAR10 dataset, which is close to random guess (10%). Consequently, in this paper, we focus primarily on how to query-efficiently train a good student model from black-box models with hard labels, which is very practical but challenging.

For this purpose, we propose a novel method called *query-effIcient Data-free lEarning from blAck-box modeLs* (IDEAL), which trains the student model with two stages: a data generation stage and a model distillation stage. Instead of utilizing the teacher model (as in previous methods Truong et al. (2021b)), we propose to adopt the student model to train the generator in the first stage, which can solve the hard-label issue and largely reduce the number of queries to the teacher model. In the second stage, we train a student model that has similar predictions as the teacher model on the synthetic samples. As a result, IDEAL requires a much less query budget than previous methods, which saves a lot of money and becomes more practical in reality.

In summary, our main contributions include:

- *New Problem:* We focus on how to query-efficiently train a good student model from black-box models with only hard labels. To the best of our knowledge, our setting is the most practical and challenging to date.

- *More Efficient:* We propose a novel method called IDEAL, which does not require any query in the data generation stage and queries the teacher only once for each sample in the distillation stage. Thus IDEAL can train a high-performance student with a small number of queries.

---

[1]https://cloud.google.com/bigquery
[2]The detailed settings can be found in Section 4.1.

- *SOTA Results:* Extensive experiments on various real-world datasets demonstrate the efficacy of our proposed IDEAL. For instance, IDEAL can improve the performance of the best baseline method (DFME) by 33.46% on MNIST dataset.

## 2 RELATED WORKS

### 2.1 WHITE-BOX DATA-FREE KNOWLEDGE DISTILLATION

Recent advances in data-free knowledge distillation have enabled the compression of large neural networks into smaller networks without using any real data Micaelli & Storkey (2019); Fang et al. (2021b); Yin et al. (2020); Chen et al. (2019); Bhardwaj et al. (2019); Haroush et al. (2020); Yoo et al. (2019); Zhang et al.. Nevertheless, all of these approaches require access to the white-box teacher model and the logits (or probabilities) calculated by the teacher model, which is not always possible in realistic scenarios. For example, according to these approaches Fang et al. (2021b); Yin et al. (2020); Chen et al. (2019); Ye et al. (2020); Xu et al. (2020), the pre-trained teacher model is regarded as a discriminator, and then the generator is adversarially trained. As the teacher model is only accessible as a black-box model, it is not possible to propagate gradients in this manner. Furthermore, the widely used KL divergence is inapplicable, as the logits of the teacher model are not accessible. Additionally, some works utilize the specific structural information of the white-box model, which also violate the black-box rules. For example, DeepIn Yin et al. (2020) used the running average statistics stored in the BatchNorm layers, while DAFL Chen et al. (2019) proposed to use features extracted by convolution filters. As a result of the above irrationality, these methods perform poorly in our settings. In more practical and challenging setting we considered in this work: data-free knowledge distillation from black-box models with only hard labels.

### 2.2 DISTILLATION-BASED BLACK-BOX ATTACKS

Previous studies have explored several ways to attack black-box models without real-world training data, which can be roughly divided into transfer-based adversarial attack Zhou et al. (2020a); Yu & Sun (2022); Wang et al. (2021b) and data-free model extraction attack Truong et al. (2021a); Kariyappa et al. (2021). Essentially, these methods are based on data-free black-box model distillation. Actually, they are designed to train substitute models in black-box situations to attack the victim model. While these methods are suitable for black-box scenarios, they mainly rely on a score-based teacher which outputs class probabilities. By contrast, our study considers a much more challenging scenario, in which a black-box teacher only returns the top-1 class. Moreover, in real-world scenarios, these black-box models usually charge for each query. To achieve good performance, these methods require millions of queries, which consume a lot of computing resources and money in real-world scenarios.

### 2.3 COMPARISON WITH RELATED WORKS

The most related work is ZSDB3 Wang (2021), which also studied data-free black-box distillation with hard labels. It proposes to generate pseudo samples distinguished by the teacher's decision boundaries and then reconstruct the soft labels for distillation. More specifically, it calculates the minimal $\ell_2$-norm distance between the current sample and those of other classes (measured by the teacher model) and uses the zeroth-order optimization method to estimate the gradient of the teacher model, which requires a large number of queries, making ZSBD3 not practical in real-world scenarios. By contrast, we consider a more challenging and practical setting where only a very small number of queries (to the teacher model) is allowed, *i.e.*, query efficient. For example, ZSDB3 requires about 1000 queries to reconstruct the soft label (logits) of a single sample on MNIST dataset, while our method only requires one query, which hugely reduces the number of queries by $1000\times$.

## 3 MAIN METHOD

### 3.1 NOTATIONS

We use $\mathcal{G}$, $\mathcal{S}$, and $\mathcal{T}$ to denote the generator, the student model, and the teacher model, respectively. $\theta_\mathcal{G}$ and $\theta_\mathcal{S}$ denote the parameters of generator $\mathcal{G}$ and student model $\mathcal{S}$, respectively. $\hat{x}$ and $\hat{y}$ denote

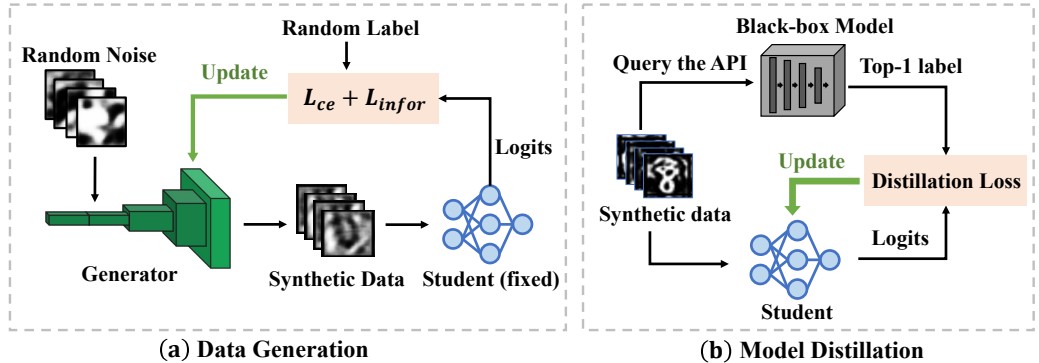

(a) **Data Generation**                    (b) **Model Distillation**

Figure 1: Illustration of the training process of our proposed IDEAL. The left panel demonstrates the data generation stage. In this stage, we train a generator, that can generate desired synthetic samples, with the student model. The right panel shows the model distillation stage, which trains a student model that has similar predictions as the teacher model on the synthetic samples.

the synthetic sample (generated by $\mathcal{G}$) and the corresponding prediction score. Subscript $i$ denotes the $i$-th sample, *e.g.*, $\hat{x}_i$ denotes the $i$-th synthetic sample. Superscript $j$ denotes the $j$-th epoch, *e.g.*, $D^j$ denotes the set of synthetic samples generated in $j$-th epoch. $C$ is the number of classes and $B$ is the batch size. We use $(\hat{y})_k$ to denote the $k$-th element of outputs $\hat{y}$, *i.e.*, the prediction score of the $k$-th class.

### 3.2 OVERVIEW

In *data-free black-box* KD, the teacher's model and data are not accessible, and we are only given the prediction of a sample by the teacher model. In particular, we focus on a more practical and challenging setting where only a category index for each sample (*i.e.*, *hard-label*) is given by the teacher model. Since each query to the teacher costs money, we consider the scenario with limited queries, *i.e.*, *query-efficient*. Our goal is to query-efficiently learn from black-box models to train a good student without any real data.

To achieve this goal, we propose a novel method called IDEAL, which consists of two stages: a data generation stage and a model distillation stage. In the first stage, instead of utilizing the teacher model (as in previous methods Truong et al. (2021b); Kariyappa et al. (2021)), we propose to adopt the student model to train the generator, which can solve the hard-label issue and largely reduce the number of queries to the teacher model. In the second stage, we utilize the teacher model and synthetic samples to train the student model. The generator and student model are iteratively trained for $E$ epochs. The training procedure is demonstrated in the Appendix (see Algorithm 1) and the illustration of the training process of IDEAL is shown in Fig. 1.

### 3.3 DATA GENERATION

In *data-free* setting, we are unable to access the original training data for training the student model. Therefore, in the first stage, we aim to train a generator to generate the desired synthetic data (to train the student model). According to the finding in Zhang et al. (2022), we reinitialize the generator at each epoch. The data generation procedure is illustrated in Fig. 1(a).

The first step is generating the synthetic sample. Given a random noise $z$ (sampled from a standard Gaussian distribution) and a corresponding random one-hot label $y$ (sampled from a uniform distribution), the generator $\mathcal{G}$ aims to generate a desired synthetic sample $\hat{x}$ corresponding to label $y$. Specifically, we feed $z$ into the generator $\mathcal{G}$ and compute the synthetic sample as follows:

$$\hat{x} = \mathcal{G}(z; \theta_{\mathcal{G}}), \tag{1}$$

where $\theta_{\mathcal{G}}$ is the parameter of $\mathcal{G}$. The synthetic samples are used to train $\mathcal{G}$.

In the second step, we compute the prediction score of $\hat{x}$. A straightforward way is to use the teacher model to compute the prediction score. The prediction score is used to update $\theta_{\mathcal{G}}$, but the parameter

of the teacher model is not accessible in *black-box* setting, thus unable to conduct backpropagation. Previous black-box KD methods Truong et al. (2021b); Wang (2021); Kariyappa et al. (2021) used gradient estimation methods to obtain an approximate gradient. Nevertheless, they need to estimate the gradient from the black-box teacher model, which requires *a large number of queries* (to the teacher model), which is not practical. Moreover, in the *hard-label* setting, the prediction score is not accessible. To this end, we propose to use the student model (instead of the teacher model) to compute the prediction score of $\hat{x}$. The detail of the student model is discussed in Section 3.4. Note that in this stage, we do not train the student model and keep the parameter of the student model fixed. By utilizing the student model, we can directly conduct backpropagation and compute the gradient of the model without querying the teacher model. In this way, we can avoid the hard-label problem and the large number of queries at the same time. The prediction score is computed as follows:

$$\hat{y} = \mathcal{S}(\hat{x}; \theta_{\mathcal{S}}), \tag{2}$$

where $S$ and $\theta_{\mathcal{S}}$ are student model and model parameters.

The third step is optimizing the generator. We propose to train a generator that considers both *confidence* and *balancing*.

### 3.3.1 CONFIDENCE

First, we need to consider confidence, *i.e.*, the synthetic sample is classified to the specified class with high confidence. To achieve this goal, we minimize the difference between the prediction score $\hat{y}$ and the specified label $y$:

$$\mathcal{L}_{ce} = CE(\hat{y}, y), \tag{3}$$

where $CE(\cdot, \cdot)$ is the cross-entropy (CE) loss. Actually, in the training process, the generator $\mathcal{G}$ can quickly converge when using $\mathcal{L}_{ce}$. Since the generated data $\hat{x}$ is fitted for the student $\mathcal{S}$, and we intend to generate data according to the knowledge of the teacher's model, we must avoid overfitting to $\mathcal{S}$. Therefore, we need to control the number of iterations $E_{\mathcal{G}}$ in data generation. Too few iterations may lead to poor data, while too many iterations may lead to overfitting. See the detailed experiments in the Section 4.1.

### 3.3.2 BALANCING

Second, we need to consider balancing, *i.e.*, the number of synthetic samples in each class should be balanced. Although we uniformly sample the specified label $y$, we observe that the prediction score $\hat{y}$ is not balanced, *i.e.*, the prediction score is high on some classes but low on the other classes. This leads to class imbalance of the generated synthetic samples. Motivated by Chen et al. (2019), we employ the information entropy loss to measure the class balance of the synthetic samples. In particular, given a batch of synthetic samples $\{\hat{x}_i\}_{i=1}^{B}$ and corresponding prediction scores $\{\hat{y}_i\}_{i=1}^{B}$, where $B$ is the batch size, we first compute the average of the prediction scores as follows:

$$\hat{y}_{avg} = \frac{1}{B} \sum_{i=1}^{B} \hat{y}_i. \tag{4}$$

Then, we compute the information entropy loss as follows:

$$\mathcal{L}_{info} = \frac{1}{C} \sum_{k=1}^{C} (\hat{y}_{avg})_k \log((\hat{y}_{avg})_k), \tag{5}$$

where $(\hat{y}_{avg})_k$ is the $k$-th element of $\hat{y}_{avg}$, *i.e.*, the average prediction score of the $k$-th class. When $\mathcal{L}_{info}$ takes the minimum, each element in $\hat{y}_{avg}$ would equal to $\frac{1}{C}$, which implies that $\mathcal{G}$ can generate synthetic samples of each class with an equal probability.

By combining the above losses, we can obtain the generator loss as follows:

$$\mathcal{L}_{gen} = \mathcal{L}_{ce} + \lambda \mathcal{L}_{info}, \tag{6}$$

where $\lambda$ is the scaling factor. By minimizing $\mathcal{L}_{gen}$, we train a generator that generates desired balanced synthetic samples.

### 3.4 MODEL DISTILLATION

In the second stage, we train the student model $\mathcal{S}$ with teacher model $\mathcal{T}$ and the synthetic samples. The training process is illustrated in Fig. 1(b). Our goal is to obtain a student model $\mathcal{S}$ that has the same predictions as teacher model $\mathcal{T}$ on the synthetic samples (generated by generator $\mathcal{G}$).

In particular, we first sample the random noise and generate synthetic sample $\hat{x}$ with the generator. Second, we feed $\hat{x}$ into the black-box teacher model and obtain its label as follows:

$$y_{\mathcal{T}} = \mathcal{T}(\hat{x}) \tag{7}$$

We treat $y_{\mathcal{T}}$ as the ground-truth label of $\hat{x}$. Since the teacher model only returns hard-label, $y_{\mathcal{T}}$ is a ground-truth one-hot label. Afterwards, we feed $\hat{x}$ into the student model and obtain the prediction score as follows:

$$\hat{y} = \mathcal{S}(\hat{x}; \theta_{\mathcal{S}}). \tag{8}$$

Last, we optimize the student model by minimizing the CE loss as follows:

$$\mathcal{L}_{md} = CE(\hat{y}, y_{\mathcal{T}}) \tag{9}$$

By minimizing $\mathcal{L}_{md}$, the student model can have similar predictions as the teacher model on the synthetic samples, which leads to a desired student model.

In each epoch, our proposed IDEAL only queries the teacher model once for each sample, while previous black-box methods (*e.g.*, ZSDB3 Wang (2021) and DFME Truong et al. (2021a)) may need more than 100 queries. Moreover, IDEAL requires similar epochs (compared with previous black-box methods) to converge. As a result, IDEAL requires a much less query budget than previous methods, which saves a lot of money and becomes more practical in reality.

## 4 EXPERIMENTS

### 4.1 EXPERIMENTAL SETUP

#### 4.1.1 DATASET AND MODEL ARCHITECTURE

Our experiments are conducted on 7 real-world datasets: MNIST LeCun et al. (1998), Fashion-MNIST (FMNIST) Xiao et al. (2017), CIFAR10 and CIFAR100 Krizhevsky et al. (2009), SVHN Netzer et al. (2011), Tiny-ImageNet Le & Yang (2015), and ImageNet subset Deng et al. (2009). The ImageNet subset is generated by Li et al. (2021), which consists of 12 classes. We resize the original image with size 224*224*3 to 64*64*3 for fast training. Note that student models cannot access to any raw data during training. Only teacher models are trained on these datasets. In this work, we study the effectiveness of our method on several network architectures, including MLP Ruck et al. (1990), AlexNet Krizhevsky et al. (2012), LeNet Lecun et al. (1998), ResNet-18 He et al. (2016), VGG-16 Simonyan & Zisserman (2015), and ResNet-34 He et al. (2016). For each dataset, we train several different teacher models to evaluate the effectiveness of our method. We use the generator proposed in StyleGAN Karras et al. (2019) as the default generator.

#### 4.1.2 BASELINES

As discussed in the related work, we compare our approach with the following baselines: 1) SOTA data-free distillation methods that are originally designed for white-box scenarios (DAFL Chen et al. (2019), ZSKT Micaelli & Storkey (2019), DeepIn Yin et al. (2020), CMI Fang et al. (2021b)). Here we adapt them to the black-box scenarios in which only hard labels are provided. 2) Besides, we also compare with the SOTA methods in model extraction attack (DFME Truong et al. (2021a)) and transfer-based adversarial attack (DaST Zhou et al. (2020a)). In fact, these techniques are essential data-free distillation methods in black-box scenarios. 3) Furthermore, we compare our method with ZSDB3 Wang (2021), which also focuses on improving the performance of the black-box data-free distillation in label-only scenarios.

Table 2: Accuracy (%) of student models trained with various teacher models on MNIST, FMNIST, SVHN, CIFAR10, and ImageNet subset. Best results are in bold. Best results of the baselines are underlined. "Improvement" denotes the improvements of IDEAL compared with the best baseline.

| Dataset | Model | Teacher | DAFL | ZSKT | DeepIn | CMI | DaST | ZSDB3 | DFME | Ours | Improvement |
|---|---|---|---|---|---|---|---|---|---|---|---|
| MNIST | MLP | 98.25 | 16.97 | 13.84 | 16.68 | 13.89 | 15.62 | 30.13 | _56.32_ | **88.41** | 32.09↑ |
| | LeNet | 99.27 | 18.92 | 22.96 | 24.43 | 22.71 | 22.49 | 35.98 | _62.86_ | **96.32** | 33.46↑ |
| | AlexNet | 99.35 | 20.43 | 27.97 | 29.54 | 28.87 | 23.86 | 37.33 | _66.45_ | **96.51** | 30.06↑ |
| FMNIST | MLP | 84.54 | 14.23 | 16.86 | 12.24 | 10.30 | 12.93 | 24.52 | _52.29_ | **76.95** | 24.66↑ |
| | LeNet | 90.23 | 16.89 | 18.52 | 16.89 | 14.44 | 22.72 | 32.46 | _56.76_ | **83.92** | 27.16↑ |
| | AlexNet | 92.66 | 21.78 | 20.22 | 22.38 | 21.24 | 25.21 | 34.47 | _63.59_ | **86.14** | 22.55↑ |
| SVHN | AlexNet | 89.82 | 16.89 | 13.96 | 16.71 | 17.63 | 24.47 | 33.96 | _58.92_ | **84.42** | 25.50↑ |
| | VGG-16 | 94.41 | 19.24 | 21.03 | 24.65 | 24.55 | 25.17 | 36.35 | _62.53_ | **86.91** | 24.38↑ |
| | ResNet-18 | 95.28 | 21.25 | 20.95 | 24.75 | 28.55 | 24.33 | 37.40 | _64.82_ | **87.65** | 22.83↑ |
| CIFAR10 | AlexNet | 84.76 | 13.75 | 12.56 | 14.54 | 13.98 | 14.54 | 29.38 | _35.73_ | **65.61** | 29.88↑ |
| | ResNet-34 | 93.85 | 16.08 | 14.31 | 15.99 | 15.95 | 15.41 | 32.18 | _37.91_ | **68.82** | 30.91↑ |
| ImageNet subset | AlexNet | 72.96 | 17.15 | 15.89 | 17.75 | 17.31 | 16.96 | 27.83 | _32.89_ | **53.72** | 20.83↑ |
| | VGG-16 | 78.53 | 19.36 | 20.16 | 19.66 | 22.10 | 22.03 | 29.46 | _34.65_ | **57.95** | 23.30↑ |

### 4.1.3 QUERY BUDGET AND TRAINING SETTINGS

Since we consider the limited query budget scenario, we adopt the same query budget $Q$ for all methods. In particular, we set the query budget $Q = 25K$ for MNIST, $Q = 100K$ for FMNIST and SVHN. Besides, the default query budget $Q = 250K$ for CIFAR10 and ImageNet subset. For large datasets with a large number of classes (*i.e.*, CIFAR100 and Tiny-ImageNet), we set the query budget $Q = 2M$. For our method, each sample only needs to query the teacher model once, so the total number of queries is $Q = B \times E$, where $B$ is the batch size and $E$ denotes the training epochs. To update the generator, we use the Adam Optimizer with learning rate $\eta_\mathcal{G} = 1e - 3$. To train the student model, we use the SGD optimizer with momentum=0.9 and learning rate $\eta_\mathcal{S} = 1e - 2$. We set the batch size $B = 250$ for MNIST, FMNIST, SVHN, CIFAR10, and ImageNet subset, and $B = 1000$ for CIFAR100 and Tiny-ImageNet datasets. By default, we set the number of iterations in data generation $E_\mathcal{G} = 5$ and the scaling factor $\lambda = 5$. The number of epochs $E$ is computed according to the query budget. For evaluation, We run experiments for 3 times, and report the average top-1 test accuracy.

### 4.2 EXPERIMENTAL RESULTS

### 4.2.1 PERFORMANCE COMPARISON ON SMALL DATASET

First, we show the results of different KD methods on MNIST, FMNIST, SVHN, CIFAR10, and ImageNet subset using various teacher models in Table 2. From the table, we observe that:

(1) Our proposed IDEAL outperforms all the baseline methods on all datasets. For instance, our method achieves 87.65% accuracy on SVHN dataset when the teacher model is ResNet-18, whereas the best baseline method DFME achieves only 64.82% accuracy under the same query budget. In general, IDEAL improves the performance of the best baseline by at least 20% under the same settings.

(2) The black-box teacher models trained on MNIST are much easier for the student to learn. Even with very few queries, the student model of our proposed IDEAL achieves over 96% accuracy on MNIST. We argue that this is reasonable because this task is simple for neural networks to solve, and the underlying representations are easy to learn. However, even for such a simple task, other methods cannot derive a good student model with the same small query budget. For example, when learning from the black-box AlexNet trained on MNIST, the best baseline DFME only achieves 66.45% accuracy.

(3) DAFL and ZSKT have the worst performance on all datasets. For example, the accuracy of ZSKT is only 12.56% when the teacher model is AlexNet on CIFAR10, which is close to random guess (10%). We conjecture this is because white-box KD methods are not suitable in black-box scenarios. These methods mainly depend on white-box information, such as model structure and probability or logits returned by the teacher model. Therefore, using these methods in black-box scenarios will significantly reduce their effectiveness.

Table 3: Accuracy (%) of student models on datasets with hundreds of classes. We use ResNet-18 as the default student network, and all results are tested under the same query budget $Q = 2M$.

| Dataset | Model | Teacher | DAFL | ZSKT | DeepIn | CMI | DaST | ZSDB3 | DFME | Ours | Improvement. |
|---------|-------|---------|------|------|--------|-----|------|-------|------|------|--------------|
| CIFAR100 | AlexNet | 66.24 | 9.65 | 10.19 | 14.53 | 12.30 | 14.22 | 14.38 | 16.48 | **23.85** | 7.37↑ |
| | ResNet-34 | 89.45 | 12.76 | 13.54 | 18.35 | 15.93 | 17.92 | 17.51 | 20.01 | **36.96** | 16.95↑ |
| Tiny-ImageNet | VGG-16 | 52.96 | 6.51 | 7.24 | 11.95 | 9.79 | 8.42 | 10.98 | 15.06 | **21.72** | 6.66↑ |
| | ResNet-34 | 64.53 | 9.78 | 10.35 | 15.79 | 12.82 | 11.21 | 12.16 | 17.64 | **27.95** | 10.31↑ |

### 4.2.2 PERFORMANCE COMPARISON ON LARGE DATASETS

In addition to the performance on small datasets, the performance of the black-box distillation method on large datasets deserves further investigation. Data-free knowledge distillation has historically performed poorly Zhou et al. (2020a); Chen et al. (2019) for datasets with a large number of classes (e.g. Tiny-ImageNet and CIFAR100), since it is very difficult to generate synthetic data with particularly rich class diversity. Thus, we also conduct experiments on datasets with more classes (at least 100 classes). Table 3 demonstrates the results of all methods on CIFAR100 and Tiny-ImageNet. As shown in Table 3, it is also difficult for all these methods to produce a good student model in the black-box scenario. However, our proposed IDEAL consistently achieves the best performance on these large datasets. For example, IDEAL outperforms the best baseline DFME by 16.95% on CIFAR100 with ResNet-34. When compared with other baseline methods, our model achieves significant performance improvement by a large margin of over 18%.

### 4.2.3 PERFORMANCE ON MICROSOFT AZURE

Following the settings in DaST Zhou et al. (2020a), we also conduct KD in a real-world scenario. In particular, we adopt the API provided by Microsoft Azure[3] (trained on MNIST dataset) as the teacher model and utilize LeNet Lecun et al. (1998) as the student model. As illustrated in Fig. 2, our method converges quickly and is very stable compared to other methods. Actually, our method achieves over 98% test accuracy after 10,000 queries, which implies that our proposed method is also effective and efficient for real-world APIs.

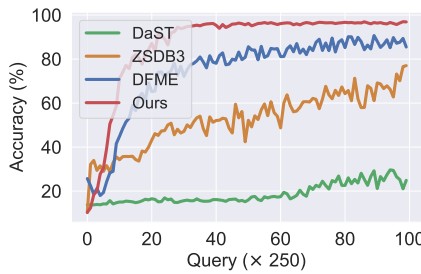

Figure 2: Transferring the knowledge of the online model on Microsoft Azure to the student.

### 4.2.4 PERFORMANCE UNDER DIFFERENT QUERY BUDGET

In previous experiments, we consider training the student model with a limited query budget. As described in previous studies Zhou et al. (2020a); Truong et al. (2021a); Wang (2021), these methods require millions of queries to the black-box model. Therefore, we have increased the number of queries of other baseline methods to provide a more comprehensive comparison, but without increasing the number of queries in our method. More specifically, we increase the number of queries required by other baseline methods (ZSDB3 and DFME) on

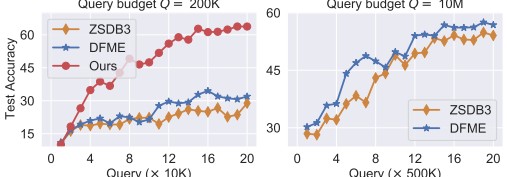

Figure 3: Analyses of our method and other comparison methods (ZSDB3, DFME) with a small query budget ($Q = 200K$) and a large query budget ($Q = 10M$).

CIFAR10 dataset from $200K$ to $10M$. Fig. 3 illustrates the training curves of these methods with $Q = 200K$ and $Q = 10M$, respectively. Note that ZSDB3, DFME can achieve the highest accuracy of 56.39% and 57.94% respectively (right panel in Fig. 3), when a large number of queries are involved. By contrast, our approach achieves 63.77% with only $0.02\times$ the query budget of both ZSDB3 and DFME. It validates the effectiveness of our method to perform query-efficient KD.

---

[3]https://azure.microsoft.com/en-us/services/machine-learning/

Table 4: Ablation studies by cutting of different modules.

| Method | MNIST | SVHN | CIFAR10 | ImageNet subset |
|---|---|---|---|---|
| Ours | **96.32** | **86.91** | **68.82** | **57.95** |
| w/o $L_{infor}$ | 95.76 | 83.21 | 62.68 | 54.31 |
| w/o $L_{ce}$ | 15.21 | 12.48 | 10.68 | 9.84 |
| w/o generator re-initializing | 95.26 | 80.53 | 55.32 | 51.25 |

### 4.2.5 VISUALIZATION OF SYNTHETIC DATA

In this subsection, we present some synthesised examples of ZSDB3, DFME, and our method to evaluate the visual diversity. As can be seen in Fig. 4, images generated by ZSDB3 are all of very low quality, which cannot show any meaningful patterns. And the image samples generated by ZSDB3 and DFME both exhibit very similar patterns, which implies that the synthetic data has low sample diversity. By contrast, our proposed approach can synthesize more meaningful and diverse data. We observe that the images generated by our method have more different patterns, which indicates that our proposed IDEAL can synthesize more diverse data. It also proves that it is feasible and effective for our model to replace $\mathcal{T}$ with $\mathcal{S}$ in generator training without gradient estimation.

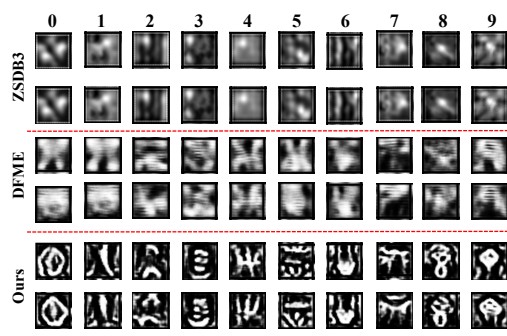

Figure 4: Visualization of data generated by different methods on MNIST. Our approach can synthesize more diverse data, there is a clear visual distinction between samples in different classes.

### 4.2.6 EFFECT INVESTIGATION OF DIFFERENT MODULES

In this section, we evaluate the contributions of different loss functions in Equation 6 used during data generation, and discuss the effect of re-initializing the generator. As shown in Table 4, removing both the generator and information loss $L_{infor}$ can lead to significant performance degradation. Moreover, our model suffers from an obvious degradation when the generator re-initializing strategy is abandoned, especially on SVHN, CIFAR10, and ImageNet-subset. In fact, since the generator is reinitialized in each epoch during training, our method does not depend on the generator from the previous round. In other words, we do not need to train the generator and the student model adversarially, and therefore we do not require a large number of training iterations to guarantee convergence. Besides, we find a significant degradation when we remove $L_{ce}$, which demonstrates its effectiveness in the data generation. The ablation experiments verify that all modules are essential in our method.

### 4.2.7 EFFECT INVESTIGATION OF $E_{\mathcal{G}}$

We also conduct ablation study to investigate the effect of $E_{\mathcal{G}}$ on the data generation stage. As show in Table 5 in Appendix, we modify the value of $E_{\mathcal{G}}$ and report the top-1 test accuracy. We can observe that too small or too large $E_{\mathcal{G}}$ is hard to obtain the optimal solution. To better understand the impact of $E_{\mathcal{G}}$, we show the t-SNE visualization of synthetic data in Fig. 5 in Appendix. More detailed results can be referred to the Appendix A.0.1.

## 5 CONCLUSION

In this paper, we propose query-effIcient Data-free lEarning from blAck-box modeLs (IDEAL) in order to query-efficiently train a good student model from black-box teacher models under the data-free and hard-label setting. To the best of our knowledge, our setting is the most practical and challenging to date. Extensive experiments on various real-world datasets show the effectiveness of our proposed IDEAL. For instance, IDEAL can improve the performance of the best baseline method DFME by 5.83% on CIFAR10 dataset with only $0.02\times$ the query budget of DFME. We envision this work as a milestone for query-efficient and data-free learning from black-box models.

## 6 ACKNOWLEDGEMENT

This work is funded by Sony AI.

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

# A APPENDIX

## A.0.1 EFFECT INVESTIGATION OF $E_\mathcal{G}$.

In Fig. 5, clearly, the student can easily identify the synthetic data when $E_\mathcal{G} = 50$ (the training accuracy on synthetic data is 100%, while the test accuracy on CIFAR10 is 58.69%), but when $E_\mathcal{G} = 10$, the student cannot distinguish the data accurately (the training accuracy is 63.78%, while the test accuracy is 68.82%). We guess that, a small value of $E_\mathcal{G}$ leads to poor quality of the generated data (a large loss) while a large value of $E_\mathcal{G}$ leads to a student model that overfits to the synthetic data. Thus, from the empirical experiments in Table 5, we set $E_\mathcal{G} = 5$ for MNIST, $E_\mathcal{G} = 10$ for CIFAR10 and ImageNet subset.

| Iterations ($E_\mathcal{G}$) | Teacher | Student | 3 | 5 | 10 | 30 | 50 |
|---|---|---|---|---|---|---|---|
| MNIST | AlexNet | LeNet | 65.36 | **96.51** | 95.24 | 88.75 | 80.15 |
| CIFAR10 | ResNet-34 | ResNet-18 | 30.25 | 57.56 | **68.82** | 62.45 | 58.69 |
| ImageNet subset | VGG-16 | ResNet-18 | 26.38 | 52.59 | **57.95** | 48.67 | 41.73 |

Table 5: The influence of different number of iterations $E_\mathcal{G}$ on data generation. We report the top-1 test accuracy (%).

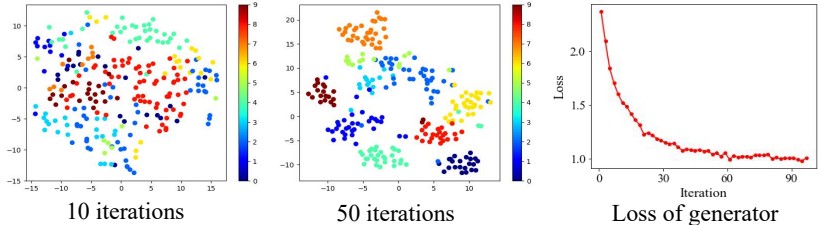

| 10 iterations | 50 iterations | Loss of generator |

Figure 5: T-SNE visualization of synthetic data on CIFAR10 and the corresponding training loss of the generator. When $E_\mathcal{G} = 10$, the features are not well separated, indicating that the student can still learn from synthetic data.

## A.0.2 EFFECT OF THE GENERATOR

For fair comparisons, we use the same generator StyleGan for all methods in our experiments. We also introduce the effects of different sizes of generators as shown in Table 6, where DCGAN, Style-GAN and Transformer-GAN have small, medium and large parameters. Different generative models have negligible effect on the performance of our method. Besides, our method still outperforms the best baseline when using generators with different sizes.

## A.0.3 CLASS IMBALANCE IN SYNTHETIC DATA

To avoid class imbalance, we generate the same number of samples per class. As illustrated in Table 7, even if we use some SOTA re-weighting methods to assign different weights to our model, the accuracy drop caused by the class imbalance can not be entirely eliminated. Hence, it is effective to consider class-balanced generation for each class, i.e., the number of synthetic samples per class is balanced.

| Generator | DCGAN | StyleGAN | Transformer-based GAN |
|---|---|---|---|
| DFME | 31.23 | 34.65 | 33.61 |
| Ours | 55.81 | 57.95 | 57.62 |

Table 6: The effect of the generator.

| Method | CIFAR10 | SVHN |
|---|---|---|
| Ours+LDAM | 62.14 | 82.24 |
| Ours+CB-Focal | 63.31 | 81.58 |
| Ours(same number of samples in each class) | **68.82** | **87.65** |

Table 7: The influence of class imbalance in synthetic data.

| Method | Model training | SVHN |
|---|---|---|
| Method_A | Train from scratch with the synthetic data | 22.64 |
| Method_B | Using out-of-domain data to distill (CIFAR10) | 39.68 |
| Ours | Using synthetic data to distill (our method) | 87.65 |

Table 8: Domain gap between synthetic data and original data

### A.0.4 DOMAIN GAP BETWEEN SYNTHETIC DATA AND ORIGINAL DATA

We find that there is a significant difference between the data synthesized by our method and the test set. As shown in Table 8, we introduce more detailed experiments to investigate such distribution discrepancy. Specifically, (1) Method_A denotes the performance when training from scratch with the synthetic data (i.e. no distillation) and directly evaluating on the test set. (2) Method_B denotes the performance of using out-of-domain CIFAR10 (i.e. no synthetic data) to perform knowledge distillation. (3) and Ours denotes the performance of using synthetic data to perform distillation.

Obviously, Ours outperforms Method_A by a large margin over 65%. It validates the significant distribution discrepancy between the synthetic data and the test set, and Ours can effectively address such domain gap via knowledge distillation. Besides, Ours performs better than domain adaptation strategy Method_B, which also verifies the effectiveness of our model for black-box.

### A.0.5 DETAILED ALGORITHM

---
**Algorithm 1** Training process of IDEAL
---
**Input:** Generator $\mathcal{G}$ with parameter $\theta_\mathcal{G}$, student model $\mathcal{S}$ with parameter $\theta_\mathcal{S}$, teacher model $\mathcal{T}$, number of training rounds $E_\mathcal{G}$ for generator in each epoch, number of classes $C$, training epochs $E$, learning rate of generator $\eta_\mathcal{G}$, learning rate of student model $\eta_\mathcal{S}$, scaling factor $\lambda$, and batch size $B$.

    **for** $e = 1, \cdots, E$ **do**
        // **Stage 1: data generation**
        **for** round $= 1, \cdots, E_\mathcal{G}$ **do**
            Sample a batch of noises and labels $\{z_i, y_i\}_{i=1}^{B}$
            Generate a batch of synthetic samples $\{\hat{x}_i\}_{i=1}^{B}$ by Equation 1
            Compute $\mathcal{L}_{gen}$ by Equation 6
            Update $\theta_\mathcal{G}$ by minimizing $\mathcal{L}_{gen}$
        **end for**

        // **Stage 2: model distillation**
        Sample a batch of noises $\{z_i\}_{i=1}^{B}$
        Generate a batch of synthetic samples $\{\hat{x}_i\}_{i=1}^{B}$ by Equation 1
        Compute $\mathcal{L}_{md}$ by Equation 9
        Update $\theta_\mathcal{S}$ by minimizing $\mathcal{L}_{md}$
    **end for**
    return $\theta_\mathcal{S}$

---

