# OpenReview forum: "IDEAL: Query-Efficient Data-Free Learning from Black-Box Models"
_ICLR.cc/2023/Conference — ICLR 2023 poster_

### Official Review · Reviewer_Ruz3 · 2022-10-23

**Confidence:** 4
**Correctness:** 4
**Technical Novelty And Significance:** 4
**Empirical Novelty And Significance:** 4
**Recommendation:** 8

**Clarity, Quality, Novelty And Reproducibility:**

Clarity: Clear enough, this presentation is clear and easy to follow.

Quality: The paper is well motivated with clear intuition and illustration. The paper is technically sound. The whole paper is well structured and easy to follow.

Novelty: Good, the main ideas of the paper are ground-breaking.

Reproducibility: good, key resources (code, data) are available and sufficient details (e.g., experimental setup) are well described.


**Strength And Weaknesses:**

Strengths：

- This paper focuses on learning from label-only black-box models, an interesting but less explored topic. From my understanding, the proposed method is by far the most practical one that considers data-free, label-only, black-box, and query-efficient learning. In Table 1, it is clear that query-efficiently training a good student model from black-box models with hard labels is very practical, but challenging. Generally, I think the investigated problem is sound and interesting. I think this can be an extremely strong paper in black-box KD.

- The idea is novel and straightforward. The approach is technically sound. The experiment showcases solid performance improvement over baselines. Particularly in Table 2, the proposed  method has significantly better results than other baselines. In Fig. 3, this paper provides convincing and solid comparisons between baselines with small and large query budgets.

-The paper also conducted very detailed and convincing ablation studies. The ablation studies are quite thorough in my view. It’s easy to understand why this method works.

Overall, this paper is very interesting and to my knowledge novel. The presentation is clear and easy to follow. Technical details are clearly described. It seems like a pioneering contribution towards black-box KD.

Weaknesses:
I didn't see major weakness in this paper actually. I am just curious about below questions:

1. How different generator sizes affect the results. Since in my opinion, synthetic data are affected by the quality of the generator. Maybe the authors can give some explanation on this and also in the future version some supplementary materials can be provided for more detailed analysis.

2. Additionally, it will be interesting to investigate why DFME and ZSDB3 perform much worse than IDEAL, what's the cause?

3. Also, is there any domain gap between synthetic data and original data? What will happen if we train the model from scratch with synthetic data?

4. Last, in the Balancing section: why it is necessary to generate the same number of samples in each class? Is there an imbalanced data distribution?


**Summary Of The Paper:**

This paper focuses on a very interesting issue: how to learn from black-box models using label-only data? For this purpose, they propose a novel method called IDEAL, which aims to query-efficiently learn from black-box model APIs in order to train a good student model without any real data. In detail, the proposed method trains the student model in two stages: data generation and model distillation. Their proposed IDEAL is query-efficient, as it does not require any query in the data generation stage and queries the teacher only once for each sample in the distillation stage. Results can well support that the proposed method consistently outperforms all the baselines.

**Summary Of The Review:**

A strong paper in black-box KD. Technical details are clearly described. It seems like a pioneering contribution towards black-box KD. Hence, I would like to vote for acceptance.

---

> ### Author Response · Authors · 2022-11-18
> **Response to Reviewer Ruz3**
>
> Thank you very much for the positive feedback and valuable comments. We hope the following clarifications can address your concerns.
>
> > How different generator sizes affect the results.
>
> For fair comparisons, we use the same generator StyleGan for all methods in our experiments.  We also introduce the effects of different sizes of generators as shown in the following table, where DCGAN[1], StyleGAN[2] and Transformer-GAN[3] have small, medium and large parameters.  Different generative models have negligible effect on the performance of our method.  Besides, our method still outperforms the best baseline when using generators with different sizes.
>
> | **Method / Generator** | **DCGAN** | **StyleGAN** | **Transformer-based GAN** |
> |:----------------------:|:---------:|:------------:|:-------------------------:|
> |          DFME          |   31.23   |     34.65    |           33.61           |
> |          Ours          |   **55.81**   |     **57.95**    |           **57.62**           |
>
>
>
> We hope the new results can adequately address your concerns.
>
> [1]Radford A, Metz L, Chintala S. Unsupervised representation learning with deep convolutional generative adversarial networks[J]. arXiv preprint arXiv:1511.06434, 2015.
>
> [2]Tero Karras, Samuli Laine, and Timo Aila. A style-based generator architecture for generative adversarial networks. CVPR 2019.
>
> [3]Jiang Y, Chang S, Wang Z. Transgan: Two transformers can make one strong gan[J]. arXiv preprint arXiv:2102.07074, 2021, 1(3).
>
> > why DFME and ZSDB3 perform much worse than IDEAL, what's the cause?
>
> Thanks for your question. First, we emphasize that our method is a query-efficient algorithm, which is one of the most significant contributions in our paper. For fair comparison, we compare all methods under the same number of queries, e.g., 25K for MNIST. For each training epoch, our method queries the black-box model once for each data. But for ZSDB3, to calculate the sample robustness of a sample, a large number of queries is required for both zero-order gradient estimation and binary searching (5000 queries for computing the sample robustness).   And for DFME, the authors recover the model’s logits from its probability predictions to approximate gradients, which also requires a large number of queries.
> As a result, these methods perform poorly in query-efficient scenarios.
>
>
> > Is there any domain gap between synthetic data and original data? What will happen if we train the model from scratch with synthetic data?
>
> Thanks for the insightful question. There is a significant difference between the data synthesized by our method and the test set. As shown in the table, we introduce more detailed experiments to investigate such distribution discrepancy. Specifically,
>
> (1) Method_A denotes the performance when training from scratch with the synthetic data (i.e. no distillation) and directly evaluating on the test set.
>
>  (2) Method_B denotes the performance of using out-of-domain CIFAR10 (i.e. no synthetic data) to perform knowledge distillation.
>
> (3) and Ours denotes the performance of using synthetic data to perform distillation.
>
> Obviously, Ours outperforms Method_A by a large margin over 65%. It validates the significant distribution discrepancy between the synthetic data and the test set, and Ours can effectively address such domain gap via knowledge distillation. Besides, Ours performs better than domain adaptation strategy Method_B, which also verifies the effectiveness of our model for black-box.
>
> | **Method** |               **Model training**              |  **SVHN** |
> |:----------:|:---------------------------------------------:|:---------:|
> |  Method_A  |   Train from scratch with the synthetic data  |   22.64   |
> |  Method_B  | Using out-of-domain data to distill (CIFAR10) |   39.68   |
> | Ours       | Using synthetic data to distill (our method)  | **87.65** |
>
>
>
> > why it is necessary to generate the same number of samples in each class? Is there an imbalanced data distribution?
>
> To avoid class imbalance, we generate the same number of samples per class. As illustrated in the table, even if we use some SOTA reweighting methods [1,2] to assign different weights to our model, the accuracy drop caused by the class imbalance can not be entirely eliminated. Hence, it is effective to consider class-balanced generation for each class, i.e., the number of synthetic samples per class is balanced.
>
> |                    Method                   |  CIFAR10  |    SVHN   |
> |:-------------------------------------------:|:---------:|:---------:|
> |                  Ours+LDAM [1]                 |   62.14   |   82.24   |
> |                Ours+CB-Focal [2]               |   63.31   |   81.58   |
> | Ours (same number of samples in each class) | **68.82** | **87.65** |
>
>
> [1] Kaidi Cao et al. Learning imbalanced datasets with label-distribution aware margin loss. In NeurIPS 2019.
>
> [2] Yin Cui et al. Class-balanced loss based on effective number of samples. In CVPR 2019

---

> ### Author Response · Authors · 2022-11-24
> **Thanks again for your valuable comments!**
>
> Dear Reviewer Ruz3,
>
> Again, we appreciate your feedback and recognition.
>
> Please also let us know if there are further questions or comments about this paper. We strive to improve the paper consistently, and it is our pleasure to have your feedback!
>
>
> Best,
>
> Authors

---

> ### Comment · Reviewer_Ruz3 · 2022-12-11
> **Thanks for the detailed response**
>
> Thanks for the detailed response, I'm satisfied with the rebuttal and the newly added results. I vote for acceptance.

---

### Official Review · Reviewer_XB73 · 2022-10-24

**Confidence:** 4
**Correctness:** 3
**Technical Novelty And Significance:** 2
**Empirical Novelty And Significance:** 3
**Recommendation:** 5

**Clarity, Quality, Novelty And Reproducibility:**

**Clarity**: Very good. The paper is easy to follow and understand. One nitpick is mis-using definitions e.g., see concern 3 (CE loss referred to as distillation loss).

**Quality**: Average. The evaluation section is quite strong and thorough (numerous KD baselines and many datasets). A concern however is better insights into the data generation step of the approach (see concern 1).

**Novelty**: Average. The paper appears to use KD as a straw-man, while it is more consistent with model extraction formulations. Moreover, the general framework (generation + student network training) is very similar to DFME, MAZE, etc. Some suggestions on improving novelty: more elaboration/technical insights into the approach (under what conditions should it work? analysis), comparisons against model extraction literature, etc.

**Reproducibility**: Very good. Although the code is not provided, I am confident in reproducing results given the implementation details in the paper.

**Strength And Weaknesses:**

### Strengths

**1. Insight: Identifying query-inefficient bottleneck**
- Although the proposed approach follows a similar approach to baselines (i.e., iterative training of generator and student model), it rightly identifies and tackles a relevant pain-point: estimating the gradients to update the generator incurs a large query budget. It appears that the drastic improvement in query efficiency can be attributed to addressing this pain-point.

**2. Results - significantly better than prior art**
- I appreciate the extensive comparison of the proposed approach with 7+ baselines and on 7+ datasets. The results are furthermore promising: the proposed approach results in drastic improvements over baselines e.g., 37.91→68.82 in CIFAR10.

### Concerns

**1. (Major concern) Counter-intuitive formulation: Generator loss**
- I found counter-intuitive how the generator is trained in stage 1. For simplicity, let's assume a perfect generator and student network. In this case, the $L_CE$ would appear to be extremely large (in expectation) -- since the GT target $y$ is randomly drawn and compared with a random generation ($\hat{y} = S(G(z)), z \sim N(0, I)$). (Side-note: wouldn't a conditional generator make more sense?)
- On a more general note, while I understand the algorithm steps, I cannot understand the reasoning behind it. Why is the gradient signal though a noisy student network with mislabeled examples informative enough to perform ZSKD?

**2. KD vs. Model Stealing approaches**
- I get the impression that the paper tackles the *Model stealing/extraction* problem (black-box teacher, query budget,
etc.), but constantly and unfairly compares to *Knowledge Distillation*.
- For instance, "KD methods are based on several unrealistic assumptions ... access teacher's training data ... white-box teacher". I don't think this is a correct claim, since KD approaches are predominantly tailored for model compression -- Hinton et al. 2015 claim that it can be used to "... compress the knowledge in an ensemble into a single model...". In which case, it's a perfectly valid assumption to use a white-box teacher and its corresponding training data.
- In contrast, the proposed approach is more consistent with model stealing works (several references missing here in the paper; please fetch and discuss relevant citations from DFME) where indeed the teacher model is a black-box prediction API. Unfortunately, the results fall short here -- randomly using a pool of publicly-available images over generated images results in significantly better accuracy scores as well as sample efficiency (typically with 50K queries, see references in Kariyappa et al., CVPR '20).
- Overall, my concern is that the paper considers KD works as a straw-man, as opposed to model extraction literature. However, I'm slightly overlooking this as the paper is "data-free" and does not rely on a publicly-available image data pool (at the expense of results degradation).

**3. Some misc. concerns**
- Table 1 "with a limited number of queries": please mention how many queries per method.
- "Contributions ... new problem ... training models with hard-labels": This has been studied plenty of times before e.g., Tramèr et al. '16, Knockoff Nets '20)
- APIs return only top-1 class: This is another unfair claim. Plenty of pay-per-query cloud APIs provide probabilities e.g., [Google Cloud](https://cloud.google.com/vision/docs/labels).
- The paper many times mentions a "distillation loss", but rather refers to simple cross-entropy given that there is no temperature-scaling factor.
- (Nitpick) Would be nice to have an additional "upper bound" column in Table 2, where the numbers reflect the teacher model trained on GT data with the relevant budget.

**Summary Of The Paper:**

- The paper proposes a zero-shot Knowledge Distillation ("ZSKD") approach (i.e., does not require any samples from teacher's training data).
- Similar to DFME (Truong et al.) and MAZE (Karyiappa et al.), the approach consists of two phases per epoch: (a) training a generator to produce synthetic labels; and (b) training a student network using generator's synthetic images annotated by a black-box teacher model. Unlike previous approaches, the proposed approach is query-efficient since gradients for the generator no longer requires 0-th order gradient estimates. Rather the gradients are generator by backprop-ing through the partially trained student network.
- Results are verified on a number of datasets (CIFAR10, TinyImages, ...) and compared with many zero-shot KD baselines.

**Summary Of The Review:**

Overall, the paper proposes a straight-forward approach with strong results (significant improvements over ZSKD baselines). However, a big concern I have is that the rationale of the approach is not evident to me (why does it work, given large losses in ideal circumstances). A somewhat minor concern I have is unfairly using KD as a straw-man (e.g., KD requires white-box teacher).

---

> ### Author Response · Authors · 2022-11-18
> **Response[1/4]**
>
> Thanks a lot for your time and effort in reviewing our paper. We appreciate it. In addition, we are pleased that you think our method solves the main pain-point of current methods, i.e., training the student model takes a large number of queries.
> Below are responses to the concerns raised by you. Please let us know if you require any further information, or if anything is unclear.
>
> > I found counter-intuitive how the generator is trained in stage 1.
>
> To begin with, we want to show that training the generator is very easy:
> In stage 1, we fix the pseudo-labels $y$ for all samples in a batch during the data generation, and the student model is also fixed, we only update the synthetic data (i.e., update the generator). The generator converges quickly after training it several epochs with this batch of data. We show the training loss curve of the generator in Figure 5 (in the Appendix), and we find that it converges rapidly within 60-90 epochs.
> Afterwards, we explored what kind of data is suitable for distillation:
>
> Although the generator converges, that does not mean the synthetic data is ideal for distillation. In an ideal situation, the data would be synthesized entirely from the knowledge of the teacher model, but this would require a large number of queries. To avoid this, we use the output of the student model to update the generator. In the training process, the generator $G$ can quickly converge, which means the synthesized data are very consistent with the student model (but not for the teacher model). Once converged, directly training the student model on synthetic data can achieve 100% accuracy in dozens of epochs. However, this does not imply that the data is well generated: while the student model can recognize the synthetic data 100%, the teacher model may produce completely inconsistent results, which means there are additional costs associated with correcting biased knowledge during the distillation process. Hence, we must avoid the synthetic data overfitting to the student model. Therefore, we need to control the number of iterations $E_G$ in data generation. As we demonstrated in Table 5 (in the Appendix), too few iterations may lead to poor data, while too many iterations may lead to overfitting. In order to distill more efficiently, we propose using the data generated within 5-10 epochs.
>
> We hope the response can adequately address your concerns.
>
> > wouldn't a conditional generator make more sense?
>
> Thanks for the suggestion. For fair comparison, we use the same generator as other methods. The table below shows our results by replacing it with conditional GAN. Experimentally, using conditional GAN does not enhance the performance of the student model. In this case, we assume that the labels $y$ are not required to participate in optimization as long as they are class-balanced. As a result, we fixed the labels of the synthetic data in each batch and made the pseudo-labels class-balanced.
>
> | **Method** | **Default Generator** | **Conditional GAN** |
> |:----------------------:|:------------:|:-------------------------:|
> |          DFME          |     34.65    |           32.61           |
> |          Ours          |      **57.95**    |           **55.62**           |
>
>
> If you are interested in whether the generated data is class-balanced, we did some additional experiments:
> To avoid class imbalance, we generate the same number of samples per class. As illustrated in the table, even if we use some SOTA reweighting methods [1,2] to assign different weights to our model, the accuracy drop caused by the class imbalance can not be entirely eliminated. Hence, it is effective to consider class-balanced generation for each class, i.e., the number of synthetic samples per class is balanced.
>
> |                    Method                   |  CIFAR10  |    SVHN   |
> |:-------------------------------------------:|:---------:|:---------:|
> |                  Ours+LDAM[1]                  |   62.14   |   82.24   |
> |                Ours+CB-Focal[2]                |   63.31   |   81.58   |
> | Ours (same number of samples in each class) | **68.82** | **87.65** |
>
> [1] Kaidi Cao et al. Learning imbalanced datasets with label-distribution aware margin loss. In NeurIPS 2019.
>
> [2] Yin Cui et al. Class-balanced loss based on effective number of samples. In CVPR 2019
>
>
> We hope the new results  can adequately address your concerns.

---

> > ### Author Response · Authors · 2022-11-18
> > **Response [2/4]**
> >
> > > Why is the gradient signal through a noisy student network with mislabeled examples informative enough to perform ZSKD?
> >
> > Note that the entire training process is conducted through a loop, i.e., stage 1 (data generation) and stage 2 (model distillation) are repeated until the student model converges:
> >
> >  (1) In stage 1, we aim to fully mine the information of the student model and generate data that matches its predictions. The student model determines the quality of synthetic data completely, and the teacher model doesn't participate in stage 1.
> >
> >  (2) In stage 2, by using the data generated in stage 1, the teacher model transfers knowledge to the student model.
> >
> > The randomly initialized student model gradually learns the knowledge of the teacher model by knowledge distillation, thereby improving the quality of the synthetic data as well. In this way, the gradient signal through a student network with generated data is informative enough to perform ZSKD.
> >
> > Related work [1] also showed that even noisy data could be used to distill certain information.
> >
> > [1] Truong J B, Maini P, Walls R J, et al. Data-free model extraction[C]//Proceedings of the IEEE/CVF Conference on Computer Vision and Pattern Recognition. 2021: 4771-4780.
> >
> > Please let us know if this answers your question.
> >
> > > I get the impression that the paper tackles the Model stealing/extraction problem (black-box teacher, query budget, etc.), but constantly and unfairly compares to Knowledge Distillation.
> >
> > In our experiments, we compare a wide variety of methods with similar backgrounds (i.e., data-free, black-box), including state-of-the-art methods in the field of model stealing. In detail, we compare our approach with the following baselines:
> >
> > (1) SOTA data-free distillation methods that are originally designed for white-box scenarios (DAFL[1], ZSKT[2], DeepIn[3], CMI[4]). Here we adapt them to the black-box scenarios in which only hard labels are provided.
> >
> > (2) Besides, we also compare with the SOTA methods in model extraction attack (DFME[5]) and transfer-based adversarial attack (DaST[6]). In fact, these techniques are essential data-free distillation methods in black-box scenarios.
> >
> > (3) Furthermore, we compare our method with ZSDB3[7], which also focuses on improving the performance of the black-box data-free distillation in label-only scenarios.
> >
> > A more detailed discussion can be found in the Related Work. We believe our experiments are convincing since we compared methods in various fields (essentially data-free & black-box KD).
> >
> > [1]Hanting Chen, Yunhe Wang, Chang Xu, Zhaohui Yang, Chuanjian Liu, Boxin Shi, Chunjing Xu, Chao Xu, and Qi Tian. Data-free learning of student networks. In Proceedings of the IEEE/CVF International Conference on Computer Vision, pp. 3514–3522, 2019.
> >
> > [2] Paul Micaelli and Amos J. Storkey. Zero-shot knowledge transfer via adversarial belief matching. In Hanna M. Wallach, Hugo Larochelle, Alina Beygelzimer, Florence d’Alche-Buc, Emily B. Fox, ´ and Roman Garnett (eds.), Advances in Neural Information Processing Systems, pp. 9547–9557, 2019.
> >
> > [3]Hongxu Yin, Pavlo Molchanov, Jose M. Alvarez, Zhizhong Li, Arun Mallya, Derek Hoiem, Niraj K. Jha, and Jan Kautz. Dreaming to distill: Data-free knowledge transfer via deepinversion. In IEEE/CVF Conference on Computer Vision and Pattern Recognition, pp. 8712–8721, 2020.
> >
> > [4]Gongfan Fang, Jie Song, Xinchao Wang, Chengchao Shen, Xingen Wang, and Mingli Song. Contrastive model inversion for data-free knowledge distillation. CoRR, abs/2105.08584, 2021b.
> >
> > [5]Jean-Baptiste Truong, Pratyush Maini, Robert J. Walls, and Nicolas Papernot. Data-free model extraction. In IEEE Conference on Computer Vision and Pattern Recognition, pp. 4771–4780, 2021a.
> >
> > [6]Mingyi Zhou, Jing Wu, Yipeng Liu, Shuaicheng Liu, and Ce Zhu. Dast: Data-free substitute training for adversarial attacks. In IEEE/CVF Conference on Computer Vision and Pattern Recognition, pp. 231–240, 2020a.
> >
> > [7]Zi Wang. Zero-shot knowledge distillation from a decision-based black-box model. In Marina Meila and Tong Zhang (eds.), Proceedings of the 38th International Conference on Machine Learning, volume 139, pp. 10675–10685, 2021.

---

> > > ### Author Response · Authors · 2022-11-18
> > > **Response [3/4]**
> > >
> > > >For instance, "KD methods are based on several unrealistic assumptions ... access teacher's training data ... white-box teacher". I don't think this is a correct claim, since KD approaches are predominantly tailored for model compression -- Hinton et al. 2015 claim that it can be used to "... compress the knowledge in an ensemble into a single model...". In which case, it's a perfectly valid assumption to use a white-box teacher and its corresponding training data.
> > >
> > > Thanks for the insightful question. The answer to this question depends on whether the KD technique is used for good or for malicious purposes. A model publisher must hope that their data and models will be protected in the event that a third party utilizes the teacher model for malicious purposes (such as model stealing [1,2] or transfer-based adversarial attacks [3,4]). In cases where a third party is trusted, it is easier to train a high-performance student model by providing a white-box model and original training data.
> > >
> > > [1]Jean-Baptiste Truong, Pratyush Maini, Robert J. Walls, and Nicolas Papernot. Data-free model extraction. In IEEE Conference on Computer Vision and Pattern Recognition, pp. 4771–4780, 2021a.
> > >
> > > [2]Sanjay Kariyappa, Atul Prakash, and Moinuddin Qureshi. Maze: Data-free model stealing attack using zeroth-order gradient estimation, 2020.
> > >
> > > [3]Mingyi Zhou, Jing Wu, Yipeng Liu, Shuaicheng Liu, and Ce Zhu. Dast: Data-free substitute training for adversarial attacks. In IEEE/CVF Conference on Computer Vision and Pattern Recognition, pp. 231–240, 2020a.
> > >
> > > [4]Mengran Yu and Shiliang Sun. Fe-dast: Fast and effective data-free substitute training for black-box adversarial attacks. Comput. Secur., 113:102555, 2022. doi: 10.1016/j.cose.2021.102555. URL https://doi.org/10.1016/j.cose.2021.102555.
> > >
> > > > please fetch and discuss relevant citations from DFME) where indeed the teacher model is a black-box prediction API.
> > >
> > > First, we would like to point out that we have compared our method with the SOTA method DFME, which achieves leading performance on data-free model stealing. For some relevant citations discussed in DFME, we provide the following discussion:
> > >
> > > - surrogate dataset: Existing model stealing attacks either use surrogate data or synthetic datasets derived from partial access to the target dataset, such as KnockoffNets[1] and JBDA[2]. Due to privacy concerns, surrogate datasets are often hard to obtain in real-world scenarios. In addition, as reported in MAZE[3], none of these methods outperform MAZE. So we did not compare our method with these methods.
> > >
> > > - data-free: MAZE[3] is another method of data-free model stealing, which is also compared in DFME. According to DFME, DFME performs better than MAZE, so we only compared our method with DFME in our experiments.
> > > If you think it would be helpful, we could add a more detailed discussion about data-free model stealing attacks in Related Work.
> > >
> > > [1]Tribhuvanesh Orekondy, Bernt Schiele, and Mario Fritz. Knockoff nets: Stealing functionality of black-box models. In Proceedings of the IEEE Conference on Computer Vision and Pattern Recognition, pages 4954–4963, 2019.
> > >
> > > [2]Nicolas Papernot, Patrick McDaniel, Ian Goodfellow, Somesh Jha, Z Berkay Celik, and Ananthram Swami. Practical black-box attacks against machine learning. In Proceedings of the 2017 ACM on Asia conference on computer and communications security, pages 506–519, 2017
> > >
> > > [3] Sanjay Kariyappa, Atul Prakash, and Moinuddin Qureshi. Maze: Data-free model stealing attack using zeroth-order gradient estimation, 2020.
> > >
> > > > Table 1 "with a limited number of queries": please mention how many queries per method.
> > >
> > > Thanks for pointing this out. For all experiments, we set the query budget $Q=25K$ for MNIST, $Q=250K$ for CIFAR10, and $Q=2M$ for CIFAR100. We have clarified it in the updated version.
> > >
> > > > "Contributions ... new problem ... training models with hard-labels": This has been studied plenty of times before e.g., Tramèr et al. '16, Knockoff Nets '20)
> > >
> > > Previous methods require a large number of queries, which are often impractical in practice. Unlike previous methods, our paper is the first to focus on **query-efficiently** training a good student model from black-box models with hard labels.

---

> > > > ### Author Response · Authors · 2022-11-18
> > > > **Response [4/4]**
> > > >
> > > > > APIs return only top-1 class: This is another unfair claim. Plenty of pay-per-query cloud APIs provide probabilities e.g., Google Cloud.
> > > >
> > > > As we discussed in the Table 1, Model APIs with probabilities are often easier to distill than those with hard labels. Both cases can be solved using our method, only the distillation loss needs to be modified, but we prefer to solve the more challenging hard-label scene.
> > > >
> > > > Besides, in real-world applications, a pre-trained model stored on the remote server may only provide APIs for inference, these APIs usually return a category index for each sample (i.e., hard-label). For example, speech recognition systems like Siri and Cortana are trained with internal datasets and only return the results to users[1]. Cloud-based object classification systems like Clarifai [2] just give the top-1 classes of the identified objects in the images uploaded by users.
> > > >
> > > > [1] López G, Quesada L, Guerrero L A. Alexa vs. Siri vs. Cortana vs. Google Assistant: a comparison of speech-based natural user interfaces[C]//International conference on applied human factors and ergonomics. Springer, Cham, 2017: 241-250.
> > > >
> > > > [2] Clarifai, I. Clarifai: Computer vision and ai enterprise platform. 2020. URL http://www.clarifai.com.
> > > >
> > > >
> > > > > The paper many times mentions a "distillation loss", but rather refers to simple cross-entropy given that there is no temperature-scaling factor.
> > > >
> > > > Yes, in our experiments, we all use the cross-entropy loss function as the loss function in the distillation stage.  It is not only the function with temperature-scaling factor that can be called the distillation loss. For example, in Dast [1] and DFME [2], the cross-entropy loss, L1 distance, and KL loss are used as the loss function for distillation.
> > > >
> > > > [1] Mingyi Zhou, Jing Wu, Yipeng Liu, Shuaicheng Liu, and Ce Zhu. Dast: Data-free substitute training for adversarial attacks. In IEEE/CVF Conference on Computer Vision and Pattern Recognition, pp. 231–240, 2020a.
> > > >
> > > > [2]Jean-Baptiste Truong, Pratyush Maini, Robert J. Walls, and Nicolas Papernot. Data-free model extraction. In IEEE Conference on Computer Vision and Pattern Recognition, pp. 4771–4780, 2021a.

---

> ### Author Response · Authors · 2022-11-24
> **With the hope that our response addresses your concerns**
>
> Dear Reviewer XB73,
>
> Thank you for your valuable time and insightful comments again!
>
> We have offered a detailed answer to your concerns. Please let us know if any of your concerns were not adequately addressed. We would be delighted to incorporate your comments into the rebuttal update.
>
> Best,
>
> Authors

---

> ### Author Response · Authors · 2022-12-04
> **Discussion Period 2**
>
> Dear Reviewer XB73,
>
> Thanks for your valuable time. Please let us know if you have any additional comments or require further clarification on any of the points raised so that we can try to address them before the end of the discussion period on December 12.
>
> Thank you in advance for your help.
>
> Best,
> Authors

---

> ### Author Response · Authors · 2022-12-08
> **A Gentle Reminder of Further Feedback**
>
> Dear Reviewer XB73,
>
> The conclusion of discussion period is closing, and we eagerly await your response. We greatly appreciate your time and effort in reviewing this paper and helping us improve it.
>
> Please help us to review our responses once again and kindly let us know whether they fully or partially address your concerns and if our explanations are in the right direction. We shall be grateful for any additional feedback you could give us.
>
> Kind Regards,
>
> Authors

---

> ### Author Response · Authors · 2022-12-10
> **A friendly reminder that the discussion stage will be closed in 2 days**
>
> Dear Reviewer XB73,
>
> Thank you again for your valuable comments. Since the discussion stage will end in 2 days, we kindly ask you to respond to our rebuttal so that we can have time to address your further concern/questions.
>
> Thanks in advance!
>
> Kind Regards,
>
> Authors

---

### Official Review · Reviewer_S4b4 · 2022-10-24

**Confidence:** 4
**Correctness:** 3
**Technical Novelty And Significance:** 3
**Empirical Novelty And Significance:** 3
**Recommendation:** 8

**Clarity, Quality, Novelty And Reproducibility:**

Please see the section above for notes on clarity, quality, novelty, and reproducibility.

**Strength And Weaknesses:**

## Strengths
* The problem of query-efficient, data-free learning from a black-box teacher that outputs only hard labels is well-motivated by real-world examples (e.g., Google BigQuery). It is a challenging problem that is of high relevance to the ML community.
* The proposed approach contains novel components compared to prior work and using the student model to generate a diverse set of synthetic data is interesting.
* Empirical evaluations on various data sets are presented that support the improved effectiveness of the proposed work. Based on the results, IDEAL improves over the state-of-the-art by more than 20% on the evaluated scenarios, which is quite impressive. This trend holds for larger data sets containing a larger number of classes as well. The hyperparameters for the experiments are reported for reproducibility.
* The authors present ablation studies that help justify the various components of the method (Table 4).
* The paper is well-written with a clear exposition overall. For example, the accompanying visualization of the data generated by different methods (Fig. 4) is quite interesting and helps understand the benefit of the method.

## Weaknesses
* The proposed two-stage approach of generating synthetic data -> distilling knowledge using generated data appears in prior work (Wang, 2021). The novelty seems to lie in the synthetic data generation stage.
* The synthetic data generation seems to be highly sensitive to the number of epochs $E_{\mathcal G}$ that the generator is trained for (see Appendix A.0.1). It has to be not too high and not too low. This is quite puzzling and the explanation in Sec. A.0.1 is not very compelling. My understanding of the synthetic data generation stage is to generate a diverse set of examples with varying (student predicted) labels. Why would training the generator for more epochs lead to “overfitting of the student?” Shouldn’t it be more conducive in generating a diverse set of synthetic data points?
* Standard deviations of the results in Sec. 4 averaged over 3 trials are not reported.
* Why is a scaling factor of $\lambda = 5$ used for the experiments? How was this choice made and how sensitive is the algorithm to this choice?
* How does the method fare in computational complexity relative to the compared approaches? My understanding is that the two stage process is done on a per-epoch basis, so it is not clear how much of a computational burden this imposes. To add to this concern, the authors mention downsizing the original images for the ImageNet subset for “fast training.”


**Summary Of The Paper:**

This paper addresses the problem of query-efficient learning from a black-box teacher model in a data-free way. This paper considers the additional constraint that only the hard labels (categorical predictions) of the teacher are available, rather than the soft-labels. The authors propose the IDEAL method for this problem which operates in two stages. First, the method uses the student model to train a generator that generates synthetic data. In the second stage, the synthetic data is used to perform hard-label distillation, i.e., train a student model so that its predictions match those of the teacher. The authors present empirical evaluations that demonstrate the query efficiency and effectiveness of the proposed approach.

**Summary Of The Review:**

This paper addresses a challenging problem that is motivated by real-world applications. The authors propose an approach that performs exceptionally well in practice compared to state-of-the-art. I have some concerns regarding the sensitivity of the method to its hyperparameters and some clarifying questions that I raised in the sections above. Overall, I lean towards acceptance and would be willing to raise my score if my concerns are adequately addressed.

---

> ### Author Response · Authors · 2022-11-18
> **Response to S4b4**
>
> Thank you for reviewing our paper, we appreciate it! We are glad that you think our paper is well-written with a clear exposition overall, and that the problem we studied is of high relevance to the ML community.
>
>
> > The novelty seems to lie in the synthetic data generation stage.
>
> Yes, the method proposed by Wang is indeed a two-stage approach, but it is not query-efficient. In addition, due to the transferability of adversarial samples, transfer-based data-free adversarial attacks are easier than data-free black-box distillation. In particular, our method offers the following advantages:
>
> In the data generation stage, we use the following techniques to maintain student model performance while reducing queries: (1) we use the output of the student model to update the generator. In this way, we do not have to query the black box model and can solve the hard-label issue. Previous methods often involved gradient estimation, which incurs a large query budget. (2) Unlike the previous method (Wang 2021), we reinitialize the generator at each epoch, there is no need to adversarially train the generator and the student model. (3) We update the generator by both confidence score and balancing score.
>
> Please let us know if this answers your question.
>
> > The synthetic data generation seems to be highly sensitive to the number of epochs that the generator is trained for.
>
> In stage 1, we fix the pseudo-labels $y$ for all samples in a batch during the data generation, and the student model is also fixed, we only update the synthetic data (i.e., update the generator). The generator converges quickly after training it several epochs with this batch of data. We show the training loss curve of the generator in Figure 5 (in the Appendix), and we find that it converges rapidly within 60-90 epochs.
> Afterwards, we explored what kind of data is suitable for distillation as follows:
>
> Although the generator converges, that does not mean the synthetic data is ideal for distillation. In an ideal situation, the data would be synthesized entirely from the knowledge of the teacher model, but this would require a large number of queries. To avoid this, we use the output of the student model to update the generator. In the training process, the generator $G$ can quickly converge, which means the synthesized data are very consistent with the student model (but not for the teacher model). Once converged, directly training the student model on synthetic data can achieve 100% accuracy in dozens of epochs. However, this does not imply that the data is well generated: while the student model can recognize the synthetic data 100%, the teacher model may produce completely inconsistent results, which means there are additional costs associated with correcting biased knowledge during the distillation process. Hence, we must avoid the synthetic data overfitting to the student model. Therefore, we need to control the number of iterations $E_G$ in data generation. As we demonstrated in Table 5 (in the Appendix), too few iterations may lead to poor data, while too many iterations may lead to overfitting. In order to distill more efficiently, we propose using the data generated within 5-10 epochs.
>
> We hope the response can adequately address your concerns.
>
>
> > Why is a scaling factor of $\lambda=5$ used for the experiments? How was this choice made and how sensitive is the algorithm to this choice?Standard deviations of the results in Sec. 4 averaged over 3 trials are not reported.
>
> My apologies for not clarifying this parameter's selection scheme. With a hyperparameter search, we find that larger $\lambda$ between 3 and 10 leads to good performance. By default, we use $\lambda=5$ in our experiments.  Considering the time constraints, these experiments will be added to the appendix. Please refer to the appendix for some other experimental results if you are interested, e.g., experiments for class imbalance, domain gap, and the effect of generators.
>
> > How does the method fare in computational complexity relative to the compared approaches?
>
> Considering we want to train a high-performance student with a limited query budget, so the number of queries is the main factor. For each method, we count the number of queries to the teacher model. For the two-stage approaches, we multiply the total number of training epochs by the number of queries in each stage. And we downsize the original images for the ImageNet subset for all methods. Thus, we believe it is fair to compare the performance of each method with a limited query budget.
>
> We hope our response has adequately addressed your concerns.

---

> > ### Comment · Reviewer_S4b4 · 2022-12-11
> > **Thank you**
> >
> > Thank you for your detailed response.
> >
> > I read all the reviews and the authors' responses. In their rebuttal, the authors have clarified and addressed my main concerns. I have raised my score accordingly.

---

> ### Author Response · Authors · 2022-11-24
> **Hope our response addresses your concerns**
>
> Dear Reviewer S4b4,
>
> We want to express our appreciation for your comments and insights again.
>
> We have provided a point-to-point response to your concerns. Please kindly let us know if you have any concerns you find not fully addressed. We would be more than happy to include your suggestions in the rebuttal update.
>
> Best,
>
> Authors

---

### Official Review · Reviewer_Epxx · 2022-10-25

**Confidence:** 3
**Correctness:** 2
**Technical Novelty And Significance:** 2
**Empirical Novelty And Significance:** 2
**Recommendation:** 5

**Clarity, Quality, Novelty And Reproducibility:**

**Clarity/Quality**
* Abstract and introduction are unclear how exactly this work differs from prior sample-efficient distillation works. Particularly, the abstract focuses mostly on problem settings, not on motivation of this work, nor high-level sketch of what they are trying to do (they just simply refer to it works with two stages (“data generation stage and queries the teacher only once ...”).
* Some terminologies are also misleading. For example, “white-box” and “black-box” refers to whether we have prior knowledge of model internals such as function classes or main assumptions. It does not mean the model class produces only the hard-labels (top1) or not (soft-labels). Hence, many classic distillation setup treats teacher models as black-box models (such as the original softmax-based distillation in Hinton’s paper [1] ).
* Minor question: What are the teacher models used in Section 4?

**Novelty**
There are mainly two lines of work that are closely related for this paper.
* [2] is a very close work that is in the same problem formulation (zero-shot, black-box, and decision base) -- hence, the problem formulation is not novel unlike what authors claim in Section 1.
* [3,4] proposes to use generators during the training and [4] particularly introduced during the distillation.
This paper is basically the combination of the two lines of work, and its novelty is in proposing the joint training of the generator that can maintain the class balance and the confidence, for the zero-shot setting. Hence, the paper does have some novelty.

[1] Hinton, Geoffrey, Oriol Vinyals, and Jeff Dean. "Distilling the knowledge in a neural network." arXiv preprint arXiv:1503.02531 2.7 (2015).
[2] Wang, Zi. "Zero-shot knowledge distillation from a decision-based black-box model." International Conference on Machine Learning. PMLR, 2021.
[3] He, Xuanli, et al. "Generate, annotate, and learn: Generative models advance self-training and knowledge distillation." (2021).
[4] Zaheer, Manzil, et al. "Teacher Guided Training: An Efficient Framework for Knowledge Transfer." arXiv preprint arXiv:2208.06825 (2022).

**Strength And Weaknesses:**

**Strength**
* Empirically strong in the particular distillation setup (no data, teacher can only provide hard-labels).
* Utilizing the generator in the data-limited distillation setup has some novelty, particularly in the joint training of the generator during the distillation process.

**Weakness**
* The impact is limited on the very specific distillation setup (data-free & hard-label setting). Moreover, the algorithm can only be applicable for the classification set-up, and does not seem to scale up when there are a large set of labels (thousands or millions).
* Authors seem to be over-claiming their impact. Particularly, I disagree with the author's claim that most KD assumes “users can directly access teacher’s training data.” Widely, KD focuses on a large set of unlabeled data because a teacher can provide pseudo-labels for them.
* Writings can be improved to focus on their core motivation, core idea, and core techniques. Please see the “clarity/quality section.” Because of this reason, many of the design choices in Section 3 feel a bit arbitrary.
* One of the baseline authors chosen, ZSDB3KD does seem to show 96.54% in their paper unlike the very low number in the current paper. Why is the number significantly different?


**Summary Of The Paper:**

Authors propose the “IDEAL” algorithm that can improve the distillation process when there is no data to be used for the distillation process, and also when the teacher provides only the hard-labels (no soft predictions over the labels). IDEAL employs a generator to generate examples on-the-fly in this data-free scenario. The generator aims to balance the class distribution and fit to one of the classes while the distillation progresses. In this particular setting (no-data and hard-labels), authors claim IDEAL can reduce the number of examples to be annotated significantly and still can achieve better performance on the given task.

**Summary Of The Review:**

As discussed above, the paper has some contributions to the community. However, the paper does seem to be limited in a very specific setting and lack the clarity/quality for the acceptance bar.

---

Post-rebuttal: After reading author's response as well as other reviewer's recommendation, I agree that the paper has some original contribution to the community (although it's limited). Hence, I updated the score accordingly.

---

> ### Author Response · Authors · 2022-11-18
> **Response [1/2]**
>
> We would like to thank you for taking the time to review our paper and for providing detailed and helpful comments! We are particularly pleased that you think the paper is empirically strong in the particular distillation setup (no data, teacher can only provide hard-labels).
> With regards to the concerns, our response is a little long as we want to make sure that we can cover each point that you raised. Please let us know if you require any further information, or if anything is unclear.
>
> > The impact is limited on the very specific distillation setup (data-free & hard-label setting).
>
> We agree that the main problem setting we consider is about distillation. However, we disagree with its limited impact. In fact, the impact could be huge to both community and industry. In real-world applications, a pre-trained model stored on the remote server may only provide APIs for inference, these APIs usually return a category index for each sample (i.e., hard-label). For example, speech recognition systems like Siri and Cortana are trained with internal datasets and only return the results to users [1]. Cloud-based object classification systems like Clarifai [2] just give the top-1 classes of the identified objects in the images uploaded by users.
>
> [1] López G, Quesada L, Guerrero L A. Alexa vs. Siri vs. Cortana vs. Google Assistant: a comparison of speech-based natural user interfaces[C]//International conference on applied human factors and ergonomics. Springer, Cham, 2017: 241-250.
>
> [2] Clarifai, I. Clarifai: Computer vision and ai enterprise platform. 2020. URL http://www.clarifai.com.
>
> We hope the above clarifications can help address your concern on impact.We also want to highlight that all the other reviewers appreciate the impact of our work, and agree that “the problem of query-efficient, data-free learning from a black-box teacher that outputs only hard labels is well-motivated by real-world examples (e.g., Google BigQuery). It is a challenging problem that is of high relevance to the ML community”, “the proposed method is by far the most practical one that considers data-free, label-only, black-box, and query-efficient learning”, “this can be an extremely strong paper in black-box KD”.
>
>
>
>
> > The algorithm can only be applicable for the classification set-up, and does not seem to scale up when there are a large set of labels (thousands or millions).
>
> Thanks for the thoughtful question. We would like to argue that our main contribution of this work is not scaling up the classes. Instead, we aim to distill knowledge under the most practical setting,  which can be reflected by below 4 words: data-free, hard-label, query-efficient, black-box.
>
> Actually, black-box KD has historically performed poorly for datasets with a larger number of classes (e.g. Tiny-ImageNet and CIFAR100), since it is very difficult to generate synthetic data with particularly rich class diversity. As shown in Table 3, it is difficult for all these baseline methods to produce a good student model in the black-box scenario. However, even under this more difficult setting, our proposed IDEAL still consistently achieves the best performance on these large datasets. Although none of these baseline methods can produce good results on large datasets, we do not think that will affect the novelty of our method, on the contrary, our method has made significant progress towards scaling up the classes without significantly degrading performance. We think that our proposed method will inspire researchers to think of ways to further explore its performance on larger datasets with labels (thousands or millions).
>
> We genuinely hope that, as the first work that conducts KD under the most practical setting, our work is not penalized by not being able to cover all the datasets.
>
> > disagree with the author's claim that most KD assumes “users can directly access teacher’s training data.” Widely, KD focuses on a large set of unlabeled data because a teacher can provide pseudo-labels for them.
>
>
> If possible, please could you clarify what you mean by “KD focuses on a large set of unlabeled data”, is there any specific reference here? If you are referring to using a large amount of unlabeled data (instead of generating synthetic data) in a data-free KD scenario, as far as we know, there is very little work. Once this is clarified, we will be happy to answer your question.

---

> > ### Author Response · Authors · 2022-11-18
> > **Response [2/2]**
> >
> > > One of the baseline authors chosen, ZSDB3KD does seem to show 96.54% in their paper unlike the very low number in the current paper.
> >
> > In the first place, we emphasize that our method is a query-efficient algorithm, which is one of the most significant contributions in our paper. For fair comparison, we compare all methods under the same number of queries, e.g., 25,000 for MNIST. For each training epoch, our method queries the black-box model once for each sample. But for ZSDB3KD, to calculate the sample robustness of a sample, a large number of queries are required for both zero-order gradient estimation and binary searching (5000 queries for computing the sample robustness).  As a result, this method performs poorly with a limited query budget.
> >
> > We hope the above clarifies your concern.
> >
> > > Abstract and introduction are unclear how exactly this work differs from prior sample-efficient distillation works.
> >
> > Thank you for your comment, if possible, please could you clarify what you mean by “sample-efficient distillation works”, is there any specific reference here? Once this is clarified, we will make it clearer.
> >
> > > Black-box does not mean the model class produces only the hard-labels (top1) or not (soft-labels).
> >
> > Generally, when a white-box model is distilled, the student model can learn from the output information from the teacher’s intermediate hidden layer and logits layer (the output of the last classification layer, the input before the softmax). But for black-box models, the model APIs usually return a category index for each sample (i.e., hard-label), rather than the logits.
> >
> > We hope the response can adequately address your concerns.
> >
> > > What are the teacher models used in Section 4?
> >
> > In Section 4, all the teacher models are marked in the caption of each table. For each experiment, multiple teacher models are used, and the default student model is shown in Table 5. We copy the table here for your reference:
> >
> > |     Dataset     |  Teacher  |  Student  | Accuracy |
> > |:---------------:|:---------:|:---------:|:--------:|
> > |      MNIST      |  AlexNet  |   LeNet   |   96.51  |
> > |     CIFAR10     | ResNet-34 | ResNet-18 |   68.82  |
> > | ImageNet subset |   VGG-16  | ResNet-18 |   57.95  |
> >
> >
> > > ZSDB3KD is a very close work that is in the same problem formulation (zero-shot, black-box, and decision base)
> >
> > We highlight again that our method is a query-efficient algorithm, which is one of the most significant contributions in our paper. As shown in Table 2 and Table 3, ZSDB3KD method performs poorly with a limited query budget, and our method consistently outperforms all the baselines.
> >
> > > This paper is basically the combination of the two lines of work, and its novelty is in proposing the joint training of the generator that can maintain the class balance and the confidence, for the zero-shot setting.
> >
> > We think that novelty does not necessarily mean that the method has to be something completely new. Effective combination with several existing technologies can also be novel, as long as the proposed method is meaningful for practical problems, such as leading to significant performance improvement. For our method, to the best of our knowledge, our setting is the most practical and challenging to date. The extensive comparison with 7+ baselines and the abundant results on 7+ datasets are furthermore promising: the proposed approach results in drastic improvements over baselines e.g., 37.91→68.82 in CIFAR10. Our method can query-efficiently train a good student model from black-box models with only hard labels.
> >
> > Please let us know if this answers your question.

---

> ### Author Response · Authors · 2022-11-24
> **With the hope that our response addresses your concerns**
>
> Dear Reviewer Epxx,
>
> As the discussion period is closing, we sincerely look forward to your feedback. The authors deeply appreciate your valuable time and efforts spent reviewing this paper and helping us improve it.
>
> It would be very much appreciated if you could once again help review our responses and let us know if these address or partially address your concerns and if our explanations are heading in the right direction.
>
> Please also let us know if there are further questions or comments about this paper. We strive to improve the paper consistently, and it is our pleasure to have your feedback!
>
>
>
> Best regards,
>
> Authors

---

> > ### Comment · Reviewer_Epxx · 2022-12-11
> > **Re: rebuttal**
> >
> > I thank authors for the response. I have other reviews as well and agree on the contribution of the paper (albeit it's limited). I'll be updating my score soon.
> >
> > Here are responses for a few specific points.
> >
> > > If possible, please could you clarify what you mean by “KD focuses on a large set of unlabeled data”, is there any specific reference here?
> >
> > It is quite common. I could easily find multiple references on this setting [1, 2, 3, 4, 5]. I respectability disagree to the authors and encourage them to update the text.
> >
> > [1] Papernot, Nicolas, et al. "Semi-supervised knowledge transfer for deep learning from private training data." arXiv preprint arXiv:1610.05755 (2016).
> > [2] Kimura, Akisato, et al. "Few-shot learning of neural networks from scratch by pseudo example optimization." arXiv preprint arXiv:1802.03039 (2018).
> > [3] Menghani, Gaurav, and Sujith Ravi. "Learning from a teacher using unlabeled data." arXiv preprint arXiv:1911.05275 (2019).
> > [4] Radosavovic, Ilija, et al. "Data distillation: Towards omni-supervised learning." Proceedings of the IEEE conference on computer vision and pattern recognition. 2018.
> > [5] https://towardsdatascience.com/distilling-bert-using-unlabeled-qa-dataset-4670085cc18
> >
> > >  please could you clarify what you mean by “sample-efficient distillation works”
> >
> > I am simply suggesting the authors to illustrate how their proposed method differ from closely related work (e.g. ZSDB3KD) in abstract/introduction. Currently, it is quite unclear what the actual novelty is from the abstract or the introduction. Specifically the three bullet points on Page 2: "New Problem" / "Efficient" / "SOTA" does not contain what this paper is actually doing (side note: I respectably disagree this paper formulates a new problem)
> >
> > > blackbox/whitebox
> >
> > I still think these terminology are misleading. Blackbox models do not necessarily mean they only provide hard labels.

---

> > > ### Author Response · Authors · 2022-12-12
> > > **Thanks for your reply**
> > >
> > > Thank you very much for checking our response and thanks again for the initial comments. They are very helpful for our improvement of the paper.
> > >
> > > > KD focuses on a large set of unlabeled data
> > >
> > > Thanks for clarifing this. However, unlabeled data for KD is not always practical for privacy concerns (data-free KD does not require any real dataset). Besides, if the teacher model is trained using CIFAR10, and the unlabeled MNIST is used for KD, there will be a larger domain gap. We'll add more comparisons in the updated version.
> > >
> > > > I am simply suggesting the authors to illustrate how their proposed method differ from closely related work (e.g. ZSDB3KD) in abstract/introduction.
> > >
> > > Thanks for your suggestions. We will make this more clear in an updated version.
> > >
> > > > Blackbox models do not necessarily mean they only provide hard labels.
> > >
> > > Yes, we never said that black-box models means they only provide hard labels. But for many API systems with relatively high security, they usually return the hard labels (If it returns probability values ​​or logits, it is much easier to distill a high-performance student model). For example, speech recognition systems like Siri and Cortana are trained with internal datasets and only return the results to users[1]. Cloud-based object classification systems like Clarifai [2] just give the top-1 classes of the identified objects in the images uploaded by users.
> > >
> > > [1] López G, Quesada L, Guerrero L A. Alexa vs. Siri vs. Cortana vs. Google Assistant: a comparison of speech-based natural user interfaces[C]//International conference on applied human factors and ergonomics. Springer, Cham, 2017: 241-250.
> > >
> > > [2] Clarifai, I. Clarifai: Computer vision and ai enterprise platform. 2020. URL http://www.clarifai.com.
> > >
> > >
> > > Please also let us know if there are further questions or comments about this paper. We strive to improve the paper consistently, and it is our pleasure to have your feedback!

---

> ### Author Response · Authors · 2022-12-04
> **Discussion Period 2**
>
> Dear Reviewer Epxx,
>
> Thanks for your valuable time. Please let us know if you have any additional comments or require further clarification on any of the points raised so that we can try to address them before the end of the discussion period on December 12.
>
> Thank you in advance for your help.
>
> Best,
> Authors

---

> ### Author Response · Authors · 2022-12-08
> **A Gentle Reminder of Further Feedback**
>
> Dear Reviewer Epxx,
>
> The conclusion of discussion period is closing, and we eagerly await your response. We greatly appreciate your time and effort in reviewing this paper and helping us improve it.
>
> Please help us to review our responses once again and kindly let us know whether they fully or partially address your concerns and if our explanations are in the right direction. We shall be grateful for any additional feedback you could give us.
>
> Kind Regards,
>
> Authors

---

> ### Author Response · Authors · 2022-12-10
> **A friendly reminder that the discussion stage will be closed in 2 days**
>
> Dear Reviewer Epxx,
>
> Thank you again for your comments. Since the discussion stage will end in 2 days, we kindly ask you to respond to our rebuttal so that we can have time to address your further concern/questions.
>
> Thanks in advance!
>
> Kind Regards,
>
> Authors

---

### Decision · Program_Chairs · 2023-01-20

**Decision:**

Accept: poster

**Justification For Why Not Higher Score:**

- Novelty is limited as utilization of generative models has been explored in the literature
- Training formulation seems problematic, e.g. why generator is trained with random labels
- No comparison to baselines with random publicly-available images over generated images

**Justification For Why Not Lower Score:**

- Good empirical performance compared to considered baselines

**Metareview: Summary, Strengths And Weaknesses:**

The paper attempts to improve distillation or model extraction by reducing queries to the teacher model. It is a very important to reduce query complexity with large teachers becoming prevalent. In this regards, the authors propose a two phase approach: 1) using the current student model as a discriminator to train a generator that generates synthetic data, 2) the generated synthetic data is used to perform distillation from the teacher model. Empirical studies are conducted to show improvement in accuracy of student model for same teacher query budget over prior generative methods. There was a wide variance among reviewer scores for the paper and we thank authors and reviewers to engage in discussion towards improving the paper. Some reviewers had genuine concerns about novelty and training formulation, which remain unanswered after the author discussion. The claims (new problem ...) should be toned down as this problem has been well explored in literature both in distillation and model extraction community (also the reviewers pointed many particularly relevant citations).  Training generator with random label needs more explanation, e.g. as noise $z$ and label $y$ are sampled independently, very similar generated images might be assigned different labels and then it is not clear how generator is learning. Finally, to cover the whole landscape, which would be beneficial for the reader to know what options are available, there should be comparison to baselines using random publicly-available unlabeled images instead of generated ones. It can be clearly marked that such methods make additional assumption of availability of such data but can results in significantly better accuracy scores for the similar query budget (e.g. Kariyappa et al., CVPR '20).

**Note From Pc:**

if the above contains the word "oral" or "spotlight" please see: "oral" presentation means -> notable-top-5% and "spotlight" means -> notable-top-25%. As stated in our emails, we are disassociating presentation type from AC recommendations

**Summary Of Ac-Reviewer Meeting:**

- Reviewers maintained their positions.
- Concern about novelty and over-claiming by 2 reviewers: Authors didn't directly respond to this question posed by reviewer Epxx or the papers pointed out by the reviewer. Response to reviewer XB73 doesn't seem satisfactory either. But two positive reviewers identify subtle difference from prior works and continue to consider the paper to have valuable contributions to the community.
- Concern about training formulation: Authors tried conditional generative model as suggested by the reviewer, but its performance was worse. As both original reported result and this new result is counter-intuitive it is worth more exploration/explanation.